# Fed-Batch Fermentation of *Saccharomyces pastorianus* with High Ribonucleic Acid Yield

**DOI:** 10.3390/foods11182742

**Published:** 2022-09-07

**Authors:** Hao Chen, Jinjing Wang, Qi Li, Xin Xu, Chengtuo Niu, Feiyun Zheng, Chunfeng Liu

**Affiliations:** 1Key Laboratory of Industrial Biotechnology, Ministry of Education, School of Biotechnology, Jiangnan University, Wuxi 214122, China; 2Laboratory of Brewing Science and Technology, School of Biotechnology, Jiangnan University, Wuxi 214122, China

**Keywords:** ribonucleic acid, *Saccharomyces pastorianus*, ARTP, fed-batch fermentation

## Abstract

(1) Background: The degradation products of ribonucleic acid (RNA)are widely used in the food and pharmaceutical industry for their flavoring and nutritional enhancement functions. Yeast is the main source for commercial RNA production, and an efficient strain is the key to reducing production costs; (2) Methods: A mutant *Saccharomyces pastorianus* G03H8 with a high RNA yield was developed via ARTP mutagenesis and fed-batch fermentation was applied to optimize production capacity. Genome sequencing analysis was used to reveal the underlying mechanism of higher RNA production genetic differences in the preferred mutant; (3) Results: Compared with the highest RNA content of the mutant strain, G03H8 increased by 40% compared with the parental strain G03 after response surface model optimization. Meanwhile, in fed-batch fermentation, G03H8′s dry cell weight (DCW) reached 60.58 g/L in 5 L fermenter by molasses flowing and RNA production reached up to 3.58 g/L. Genome sequencing showed that the ribosome biogenesis, yeast meiosis, RNA transport, and longevity regulating pathway were closely related to the metabolism of high RNA production; (4) Conclusion: *S. pastorianus* G03H8 was developed for RNA production and had the potential to greatly reduce the cost of RNA production and shorten the fermentation cycle. This work lays the foundation for efficient RNA content using *S. pastorianus*.

## 1. Introduction

Ribonucleic acid (RNA) is an important biological macromolecule [1].Its degradation products such as growth-promoting substances and drug raw materials [2,3] are widely used in food, medicine and animal feed industries. Most commonly, RNA is used for condiments, for example, guanosine 5′-monophosphate (5′-GMP) is an active flavor enhancer and inosine 5′-monophosphate (5′-IMP) in combination with monosodium glutamate (MSG) can enhance the taste of umami [4]. During the commercial production of an RNA product, an efficient strain is always the key to improve quality and productivity while lowering the cost.

*Candida tropicalis* is widely used in industries to produce RNA for its higher RNA content [5,6] but with low safety [7]. Research has focused on developing *Saccharomyces cerevisiae* that produces a large amount of RNA [8,9]. Brewer’s yeast is another source for RNA production [10] for its safety [11], easy to cultivate [12], and high yield of RNA. Studies have been carried out to improve RNA production in yeast through genetic engineering by overexpressing FHL1, IFH1 and SSF2 and deleting HRP1, which plays a fundamental role in cell regulation and growth [10]. Additionally, reintroducing the essential rRNA transcription regulator RRN5, a component of the upstream activating factor to yeast strains, could also promote RNA synthesis [12]. However, this would lead to a GMO (Genetically modified organism) problem in the food industry, which does not comply with safety regulations. Traditional mutagenesis strategies, including chemical mutagenesis and physical mutagenesis, are still effective methods for strain development in the food industry. Atmospheric and room temperature plasma (ARTP) mutagenesis is a microbial mutagenesis tool based on the radio frequency (RF) glow discharge of atmospheric pressure [13]. Compared to chemical or UV mutagenesis, it is a safer, simpler and a more effective method that can induce a rich variety of DNA damage, resulting in a large library of mutations [14]. For example, a Corynebacterium glutamicum strain could produce 34.78 g/L of L-serine, which is 35.9% higher than those of the parent strain after ARTP mutagenesis, respectively [15].

The response surface methodology (RSM) has proven to be an effective method in medium optimization [6]. Fed-batch fermentation is currently an essential process to optimize fermentation and is worth researching as it could help further improve productivity and reduce costs [16,17]. A substantial increase in target product and cellular biomass can be achieved via fed-batch fermentation [18]. The production of glutathione in *S. cerevisiae* increased to 253% by continuously feeding cysteine during fed-batch fermentation [19]. Higher biomass and RNA production in *S. cerevisiae* and *C. tropicalis* were achieved with the fed-batch fermentation strategy [20,21].

In this study, ARTP mutagenesis was conducted on the *S. pastorianus* strain to develop a mutant yeast strain with a higher RNA yield. RSM and fed-batch fermentation were further applied to improve RNA production. Furthermore, genomic sequencing analysis helps us understand the mechanism for higher RNA accumulation in a cell.

## 2. Materials and Methods

### 2.1. Strains and Culture Medium

In previous studies, several yeast strains commonly used in the food industry were evaluated for RNA production and *S. pastorianus* G03 obtained relatively higher RNA content in the cell (Appendix A). G03, stored in the authors’ laboratory, was used as the parental strain, which is presented by a brewery (Jiangsu, China) and a Frohberg type lager yeast strain used for industrial beer brewing. Strains were cultured in YPD liquid medium at 28 °C, 180 r/min for 24 h and 2% agar was added in solid medium when necessary.

YPD solid medium containing 1% potassium chloride (KCl) were used to select mutants with high RNA yield [22]. Additionally, different carbon (glucose, maltose, sucrose, lactose, soluble starch, galactose, fructose, trehalose, dextrin) or nitrogen sources (peptone, urea, soy peptone, glutamine, NH_4_CI, (NH_4_)_2_SO_4_, (NH_4_)_2_C_2_O_4_, tryptone) added in YPD medium were used to observe the growth of the strain under different medium conditions. In the tolerance experiment, there were 10, 15, 20, 30, 40, 50, 60 and 70% (*w*/*w*) glucose, 4, 6, 8, 10, 12, 14, 15 and 16% (*w*/*w*) potassium chloride, 2, 4, 6, 8, 10, 12, 14 and 16% (*v*/*v*) ethanol contained in YPD medium or with a different pH (1.5, 2, 2.5, 3, 3.5, 4, 4.5, 5).

### 2.2. Screening of Mutants

Target strains were screened after several rounds’ selections, as shown in Figure 1. Mutants were obtained using ARTP mutagenesis with slight modification [23]. The helium-based ARTP mutation breeding system (ARTP-3) was from Si Qing Yuan Biotechnology Co. Ltd. (Beijing, China). Yeast cells were cultured to logarithmic phase, and the cells were washed with normal saline (0.9% NaCl) and adjusted to 10^7^ CFU/mL by cell counter (Countstar Bio Tech, Shanghai, China). The output power was 100 W, and the flow rate of the carrier gas (helium) was 10.0 L/min. The mutagenic exposure time was set at 0 to 80 s. The mutants with high RNA yield obtained in each round of mutagenesis were used as the parents for the next round mutation. The fatality rate was calculated with Equation (1) (*A* was the colony number of the control and *A_m_* was the colony number after ARTP treatment).
(1)$$\mathrm{The}\;\mathrm{fatality}\;\mathrm{rate}=\frac{A-A_{m}}{A}\times 100\%$$

By measuring the RNA content vs. the biomass of the mutants, we were able to discriminate the candidates and confirm the target mutants. The genetic stability of the mutants was measured by analyzing the changes in RNA content after eight generations’ culture. RNA of yeast cell were measured with Sasano’s methods [12]. After centrifugation of the cell culture (12,000 rpm, 30 s), the supernatant was removed and 1 mL of 0.5 N PCA was added and resuspended. After incubation at 70 °C for 20 min, cells were separated by centrifugation (12,000 rpm, 2 min) and supernatant was measured at 260 nm (*A*_260_). Dry cell weight (DCW) of 1 mL culture was measured after drying the cells at 100 ± 5 °C to meet constant weight. Total RNA content was calculated by the following equation:(2)$$\;\mathrm{RNA}\;\left(\mathrm{mg}/g\;\mathrm{DCW}\right)=\frac{A_{260}\times 0.0368}{m}$$

### 2.3. Physiological Characteristics Analysis

The fermentation ability of the strains were studied with the Duchenne tubule method [24]. Gas production reflecting the fermentation ability of the strain was observed and recorded at 4, 6, 8, and 10 h in tubules. The growth conditions of the strains in different carbon and nitrogen sources medium were measured to evaluate the strains’ ability to utilize different carbon and nitrogen sources. Yeast strains were inoculated in 20 mL of nitrogen-free YPD medium and cultured at 28 °C, 180 r/min for 48 h to prevent false positive results. Starvation-treated cell cultures in 1 mL were inoculated into 40 mL of culture medium with different carbon and nitrogen sources and cultured at 28 °C, 180 r/min for 14 h. Then, OD_600_ was measured, and the growth conditions of the cells were measured in different mediums containing different levels of tolerant parameters including glucose, KCl, ethanol or different pH.

The contents of 5’-nucleotides (5′-GMP and 5′-IMP) were determined by high performance liquid chromatography (HPLC) with modifications [25]. RNA was first extracted from yeast cells by method of RNA content measurement, and phosphodiesterase was added to carry out the enzymatic hydrolysis reaction at 70 °C for 4 h. After centrifugation of the cell culture at 5000 rpm for 4 min, and the supernatant was collected and filtered through filters (0.45 μm). The chromatographic conditions were as follows.

Column: Waters XSELECT^TM^ HSS T3 5 μm 4.6 × 250 mm Column.

Mobile phase: A—20 mmol/L KH_2_PO_4_, adjusted to pH 5.8 with phosphoric acid; B—Methanol.

Gradient: 0–10 min 100% A, 10–15 min B increases to 5%, 15–15.1 min was the same, 15.1–20 min 100% A.

The nucleotides were quantified using external calibration curves. A mixed nucleotide calibration standard solution ranging from 0 to 25 μg/mL (GMP and IMP) was prepared (5′-IMP, y = 129,215x + 56,214, R^2^ = 0.9992; 5′-GMP, y = 121,634x − 22,559, R^2^ = 0.9997).

### 2.4. Genome Sequencing and Enrichment Analysis

The yeast was collected after centrifugation at 8000 rpm for 5 min, washed twice with sterile water, treated with liquid nitrogen and stored at −80 °C for genome sequencing. The samples were sent to the BGI Genomics Co., Ltd. (Shenzhen, China). The ligation products were amplified via PCR and sequenced using Illumina HiSeq 2500. The clean data (NCBI, http://www.ncbi.nlm.nih.gov/, accessed on 23 January 2022, PRJNA799665, SUB10983201) were aligned to the reference genome, *S. pastorianus.CBS.1483* genome sequence. The KEGG database (http://www.genome.jp/KEGG/, accessed on 23 January 2022) was used to analyze the enrichment of differentially expressed genes. The DAVID bioinformatics resources 6.8 database (https://david.ncifcrf.gov/home.jsp, accessed on 23 January 2022) was used for GO functional analysis of mutant genes.

### 2.5. High-Cell-Density Fermentation

The response surface methodology (RSM) was applied to optimize the fermentation medium. Plackett–Burman (PB) designs were applied to quickly screen multiple factors and identify the impact of each factor [26]. Each variable is represented at two levels, including high (+1) and low (−1), in Table 1.

The three smallest *p*-values of these factors were considered as the significant factors which were used to enhance RNA yield by Design Expert 11.0 in central composite design (CCD). The three independent factors were observed at five different levels (−1.68, −1, 0, +1, +1.68), as shown in Table 2.

In batch fermentation—conditions changes are shown in Appendix A—the seeds were cultured in YPD medium at 28 °C, 180 r/min for 14 h. The inoculum (10%, *v*/*v*) was transferred into a 5 L fermenter with 3 L optimal medium. The ventilation was 1.8 L/min at the beginning and then manually increased when the dissolved oxygen (DO) dropped below 5%. The rotating speed was increased from 180 r/min to 250 r/min, 300 r/min at 20 h and 24 h according to the dissolved oxygen (DO). The initial pH of the medium was adjusted to the optimal pH. For fed-batch fermentation, throughout the fermentation period, the pH was automatically controlled at 5.0 when RNA content was the highest in batch fermentation by using 20% *v*/*v* H_3_PO_4_ or 20 g/L of NaOH if needed. Ventilation during incubation was constantly adjusted and peanut oil was used as an antifoam agent. Molasses began to be added in the logarithmic phase (8–12 h). The concentration (c = 570 g/L) and flow rate (V = 40 mL/h) of molasses was based on the biggest and fastest consumption in batch fermentation. The flow rate was reduced to 20 mL/h when the dissolved oxygen increased. The samples were taken every 4 h to measure the reduction in sugar content, DCW and RNA production. Fermentation stopped until DO greatly increased and growth kinetic parameters’ calculations were referred to the research [27].

### 2.6. Measurement of Reducing Sugar Content

The reduction in sugar content was measured with 3,5-dinitrosalicylic acid (DNS) method [28]. The sucrose in 2 mL cultures was hydrolyzed by 500 μL HCI (*v:v* = 1:1) and the reaction was carried out for 15 min at 70 °C in a water bath. The reaction was stopped by adding 600 μL 200 g/L NaOH. Then, 540 nm (A) was measured when it cooled. Then, 500 μL of water was added to 700 µL solution and the system was tested under the same conditions, which was blank (A_0_). The standard curve was drawn using 10 g/L glucose after dilution; the glucose content was the abscissa and the absorbance value was the ordinate (y = 3.681x − 0.1664, R^2^ = 0.994). The formula for calculating the reduction in sugar content is as follows (1.55, Dilution factor):(3)$$\mathrm{Reducing}\;\mathrm{sugar}\;\mathrm{content}\;\left(g/L\right)=\left(A-A_{0}+0.1664\right)*1.55/3.681$$

### 2.7. Statistical Analysis

Statistical analysis was carried out with the standard deviation calculation using Origin Pro 2017. Differences with a *p* value < 0.05 were considered a statistically significant difference. The experiments were performed in triplicates. The data presented in the tables and figures are the average values of the triplicate experiments with the standard deviations.

## 3. Results

### 3.1. Screening of High RNA Producing Yeast

The optimum exposure time with ARTP was studied (Appendix A) and 80 s exposure was determined for mutation. Mutants which were sensitive to KCl were selected and twelve mutants were screened after six rounds of mutagenesis. Among them, the RNA content of G03H8 and G03H11 reached 129.0 mg/g DCW (Figure 2), which was 40% higher than that of G03. Meanwhile, most of the strains maintained similar RNA content after eight generations’ culture, and G03H9 showed the best genetic stability (Appendix A). However, the RNA content of G03H9 was 126.1 mg/g DCW, which was lower than that of G03H8. G03H11 showed slightly lower genetic stability than that of G03H8 after prolonged culture. Therefore, *S. pastorianus* G03H8 was selected as the optimum strain for further study.

Compared to the parental strain G03, the mutant G03H8 was found to start fermentation earlier (Appendix A) and have better assimilation ability of maltose and galactose. However, it lost the assimilation ability of urea (Appendix A) and had little change in tolerance (Appendix A). As shown in Table 3, the flavored nucleotide content of G03H8 was twice as high than that of G03, indicating G03H8 had the potential for application in the condiment industry.

### 3.2. Genomic Sequencing of Mutant Strain

Fungal resequencing can analyze the types of genomic variation such as single nucleotide polymorphisms (SNPs), insertions, and deletions between closely related strains [8]. This can provide guidance and a basis for the screening of excellent traits of strains, research on drug resistance of strains and population evolution. The results of genome resequencing as shown in Appendix A suggest that the percentage of sequences was over 98.65% mapped to the genome, which had high contrast, large gene coverage and depth of coverage, and strong differentiation.

Some genes with a significant number of differences were considered in accordance with the KEGG pathway (Figure 3). For each cluster, the functions of mutated genes concerning riboflavin metabolism and one carbon pool by folate had the highest enrich ratio. However, other metabolic pathways involved in these genes were closely related to the metabolic pathway of high nucleic acid-producing *S. cerevisiae*, such as ribosome biogenesis, yeast meiosis, RNA transport, MAPK signaling pathway, tryptophan metabolism, carbon metabolism, and longevity regulating pathway. The results showed that RNA biosynthesis is associated with cell growth. Ribosome is composed of four RNAs (>5400 nucleotides), which are all rRNA and 79 different proteins [29]. Ribosome biogenesis is closely related to intracellular RNA accumulation. Yeast meiosis is one of the ways that yeast proliferate. RNA transport is the process by which RNA molecules move from one cellular compartment or region to another cellular compartment or region [30]. The MAPK pathways have important roles in their physiology and development; e.g., cell cycle control, mating, morphogenesis, response to different stresses, and to temperature changes, cell wall assembly and integrity, degradation of cellular organelles, virulence, cell–cell signaling, and soon on [31]. Carbon metabolism might reflect the carbon source utilization ability of the strain. The longevity regulating pathway affects the cell cycle [32]; the longer the cell cycle, the more biomass accumulation, which is one of the important ways to improve yeast RNA production.

### 3.3. High-Density Fermentation Strategy

#### 3.3.1. Optimization of Culture Conditions

The optimum growth temperature for yeast growth and RNA accumulation was 28 °C (Figure 4a), where the DCW and RNA content of G03H8 both reached the highest level. The growth of G03H8 was significantly inhibited when the temperature was higher than 30 °C. Variation in pH did not seriously affect the RNA accumulation in G03H8; however, the higher pH of culture medium inhibited the growth of the yeast cells (Figure 4b). The pH value on the scale of 4.5 to 6 could be suitable for yeast culture. Considering the convenient use of raw material in industrial production, yeast extract and (NH_4_)_2_SO_4_ were selected as the nitrogen source in RSM optimization. After comprehensive consideration, molasses was chosen as the carbon source, and the yeast was cultured at 28 °C, with an initial pH 6.

#### 3.3.2. Optimization of Fermentation Medium

The amount and ratio of nutrients affect the growth and metabolism of yeast, which further affect the accumulation of target products. Therefore, the response surface method was used to optimize the culture medium for G03H8. Molasses is often used as a carbon source for microbial fermentation due to its low cost and rich composition [33]. Phosphate ion is precursor for PRPP synthesis, which is the prerequisite to the high-level production of nucleotides [34] and Cu^2+^, Fe^2+^, Zn^2+^ are very useful for RNA production [6]. Additionally, the use of inorganic nitrogen sources can effectively reduce fermentation costs. The Plackett–Burman (PB) design was employed to distinguish the importance of main nutrients including molasses, (NH_4_)_2_SO_4_, KH_2_PO_4_, yeast extract, MgSO_4_, ZnSO_4_ and FeSO_4_. A 12-run PB design was used (Appendix A), and the experimental responses were analyzed by the method of least squares to fit the following first-order model:RNA = 14.595 + 0.2191A − 0.1424B + 1.835C − 1.015D − 1.89E − 1.49F − 0.1540G(4)

The regression coefficients for the linear regression model of RNA yield (Table 4) showed the model was significant (*p* < 0.05) with R^2^ of 0.9491, indicating that 94.91% of the variability in the response could be explained by the model. Additionally, the confidence level of other variables was lower than 95%, which was considered insignificant. Based on the statistical analysis, the factors which had the greatest impact on the yield of RNA were identified as molasses, (NH_4_)_2_SO_4_ and KH_2_PO_4_. These variables were ranked as follows: molasses > (NH_4_)_2_SO_4_ > KH_2_PO_4_.

Based on regression results of PB design, the coefficient of molasses and KH_2_PO_4_ was positive, while the coefficient of (NH_4_)_2_SO_4_ was negative. This meant that increasing the concentration of molasses and KH_2_PO_4_ and decreasing the concentrations of (NH_4_)_2_SO_4_ would have positive effects on RNA yield. Therefore, these factors were considered in the experimental design of the steepest ascent (Appendix A) and the corresponding results are shown in Appendix A. The concentrations of the yeast extract, MgSO_4_, ZnSO_4_ and FeSO_4_ were fixed at zero. As the results showed, the maximum RNA yield was obtained in run 7 and this point was chosen as the zero level of the central composite design (CCD) for further optimization.

A highly significant predicted model obtained by CCD was very useful in determining the optimal concentration of the medium components which had a significant effect on RNA yield. A set of 20 experiments (Appendix A) in which each variable was studied at five coded levels (−1.68, −1, 0, +1, +1.68) was carried out by Design Expert 11.0. Additionally, regression coefficients for the CCD model, in which the *p*-value of model was 0.019, was significant, as shown in Table 5. The screened variables were expressed by the following fitted second-order polynomial equation from the regression results:RNA = 79.25 + 35.94A + 10.68B − 30.82C + 0.03AB + 18.36AC + 7.29BC − 5.08A^2^ − 13.02B^2^ − 43.43C^2^(5)

All the predicted values of the RSM model were close to the experimental values, as shown in the plot of predicted values versus experimental values (Figure 5a). Three-dimensional response surface curves were plotted to explain the interaction between two factors for maximum RNA yield (Figure 5b–d). The maximum RNA yield was obtained when the values of the test variables were as follows: molasses 4.84% (*w*/*w*), (NH_4_)_2_SO_4_ 0.62% (*w*/*w*), KH_2_PO_4_ 0.72% (*w*/*w*), MgSO_4_·7H_2_O 2% (*w*/*w*), FeSO_4_·7H_2_O 0.06 g/L, ZnSO_4_·7H_2_O 0.06 g/L and yeast extract 1%. The predicted maximum RNA yield using this model was 158.40 mg/g DCW, which has 23% improvement compared with the original culture conditions. Validation experiments were carried out under the model-predicted medium composition. The RNA content obtained from was 158.03 (mg/g DCW), which was in excellent agreement with the predicted value.

#### 3.3.3. Fed-Batch Fermentation

Scale-up fermentation is necessary for the further application of yeast strains in the industry. Fed-batch fermentation is a common strategy used in the fermentation industry. Here, in this study, fed-batch fermentation was carried out in a 5 L fermenter containing 3 L optimum culture medium. In batch fermentation (Figure 6a), it could be found that when the pH was around 5, the physiological activity of the strain was highest at 8 h. Yeast biomass began to accumulate along with the consumption of molasses, and the molasses was almost used up after 12 h. After 28 h of culture, the RNA content reached 0.55 g/L with the DCW of 6.17 g/L.

In order to obtain higher biomass during continuous fermentation, molasses was continuously fed to the fermentation at 8 h. With the rapid cell growth and consumption of substrate, the dilution rate increased gradually. On the one hand, the addition of fresh medium could maintain the concentration of the substrate and provide nutrient substance for G03H8. On the other hand, it could reduce the problem of catabolite toxicity and maintain the regular growth of the cell. The pH was automatically maintained at 5, and the temperature was set at 28 °C. The dry weight of the G03H8 reached 60.58 g/L and the RNA yield reached 3.58 g/L after 44 h of fermentation (Figure 6b).

Fed-batch fermentation showed obvious advantages over single batch fermentation. The cell biomass increased from 6.17 to 60 g/L, the utilization rate of the substrate increased from 0.26 to 1.58 g/L/h, cell yield coefficient increased from 0.19 to 0.43 g/g and the production efficiency increased from 33.33 to 75.44% in fed-batch fermentation as shown in Table 6. It showed greatly improved fed-batch fermentation. Cell growth occurred very rapidly at a high concentration of nutrient substance, and this undoubtedly affected steady-state growth and continuous fermentation. Therefore, the lower RNA and DCW concentrations might be relative to the reducing sugar content, and continuous nutrient feeding was particularly important for cell biomass accumulation. Fed-batch culture was controlled on steady-state conditions by feeding molasses to maintain biomass and nutrient substance concentration. The growth of the cell could keep constant viable counts, showing that fed-batch culture could be maintained for a much longer time than batch fermentation. Thus, fed-batch fermentation had more economic effectiveness than batch mode.

## 4. Discussion

The sources of ribonucleic acid are relatively wide; the industrial production method of ribonucleic acid is mainly the yeast fermentation method at present. *Saccharomyces cerevisiae* [8] and *Candida lipolytica* [35] are mainly used for the production of RNA. *S. cerevisiae* has priority in the food industry due to its higher safety. In previous work, *S. pastorianus* had higher RNA and cell yield when compared to other *S. cerevisiae* such as wine yeast and baker’s yeast. However, there was no targeted use of *S. pastorianus* for high RNA yield strain breeding. In this study, a mutant *S. pastorianus* G03H8 with high RNA yield was developed via ARTP mutagenesis. However, it still needed to be cultured on solid medium with a selectable property, which prolonged the selection period. A more efficient screening method or a higher RNA yield in *S. pastorianus* are the goals of scientific research, such as high-throughput screening methods [36]. In addition, the direction of mutation is uncertain in the mutation breeding technique. At present, there are many other efficient and accurate technologies in the field of yeast breeding, such as protoplast fusion technology [37], metabolic engineering technology [38], etc.

High-cell-density culture is a widely used operation model in various industries that provides a higher product yield. It can become more effective by considering multiple factors in process design and implementation, such as microbial strains, their metabolic characteristics and nutritional requirements. The factors affecting high-density culture include types of nutrients, culture conditions, culture methods, and culture systems, etc. Posada-Uribe et al. [39] increased the number of spores and the spore rate of *Bacillus subtilis* EA-CB0575 by 17.2 times and 1.9 times, respectively, under the optimal carbon source and inorganic salt culture conditions. In other respects, the pH, which was controlled to be constant during the fermentation process, was more favorable for the strain’s growth compared to no controlling [40]. The choice of culture method plays an important role in the growth of bacteria. The process of high-density fermentation production is mainly fed-batch fermentation [41]. Fed-batch fermentation is to add fresh fermentation broth during the fermentation process to maintain sufficient nutrients and promote strains’ growth [42].

In this study, a high-density fermentation strategy for molasses feeding was developed by comparing batch fermentations. Molasses contains various nutrients such as abundant sugars, amino acids, organic acids, inorganic compounds, and vitamins [33]. This might be the reason why the growth of G03H8 could be satisfied with feeding only molasses. The increase in RNA production was closely related to higher growth. However, in the late stage of fermentation, the nitrogen and phosphorus sources in the medium were lacking. The fermentation cycle would increase with the simultaneous addition of carbon, nitrogen and phosphorus sources in fed-batch fermentation. By observing the changes in the DO in the batch fermentation process, it was found that the dissolved oxygen level in the middle of the fermentation (24–36 h) remained around 5%, indicating that the dissolved oxygen in the system could not meet the physiological activity of G03H8. This showed that dissolved oxygen limited the growth of G03H8. Therefore, it is particularly important to relieve the low dissolved oxygen limitation of yeast in the process of high-cell-density fermentation. Additionally, there was no change in specific growth rate of G03H8 between batch and fed-batch fermentation. The situations could be improved with aeration equipment, controlling of nutrients’ concentration, a bioreactor [43] and culture system [44], etc.

## 5. Conclusions

G03H8, a *S. pastorianus* mutant with high RNA yield, was obtained via ARTP mutagenesis. Compared with the parental strain G03, the fermentation speed of G03H8 was faster. However, it had little effect on the assimilation ability of different carbon and nitrogen sources and environmental tolerance. Genome sequencing showed that the mutation direction was concentrated in ribosome biogenesis, yeast meiosis, RNA transport, MAPK signaling pathway, tryptophan metabolism, carbon metabolism, and longevity regulating pathway. After response surface model optimization, the highest RNA content of mutant strain G03H8 increased by 40% compared with the parental strain G03. The RNA production and DCW of G03H8 reached 3.58 g/L and 60.58 g/L in fed-batch with molasses flowing. These findings showed that G03H8 had the potential to greatly reduce the cost of RNA production and shorten the fermentation cycle and provided valuable insight into how to increase the RNA content in *S. pastorianus* strains for industrial production. The optimum culture composition obtained in these experiments and high-cell-density fermentation of molasses feeding provided a basis for further study using large-scale fermentation for RNA production.

## Figures and Tables

**Figure 1 foods-11-02742-f001:**
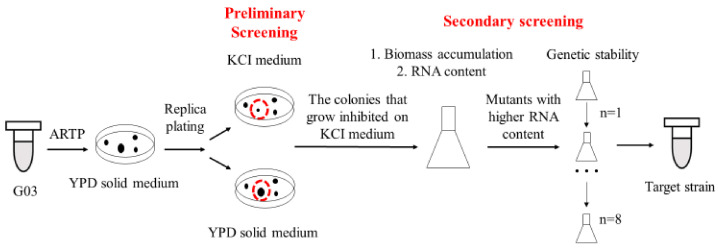
Technical route of high-ribonucleic acid mutants screening.

**Figure 2 foods-11-02742-f002:**
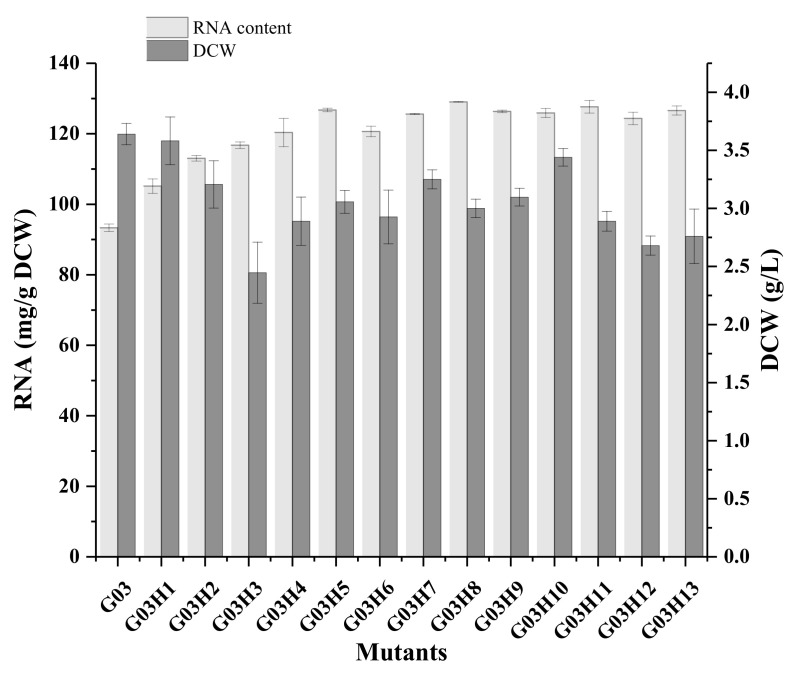
RNA content and growth of the different mutants. Data are the average of three independent experiments. Error bars represent ± SD.

**Figure 3 foods-11-02742-f003:**
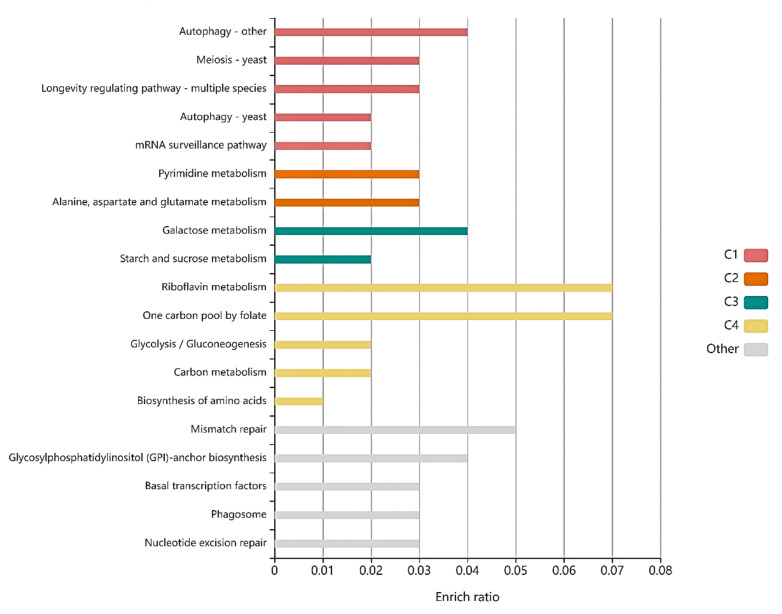
Histogram presentation of genes mapped to the KEGG database. The horizontal coordinates represent the pathway name, and the vertical coordinates represents the rich factor. The color of the bar (C1, C2, C3, C4, Other) represents different clusters. For each cluster, if there are more than 5 terms, top 5 with the highest enrich ratio will be displayed.

**Figure 4 foods-11-02742-f004:**
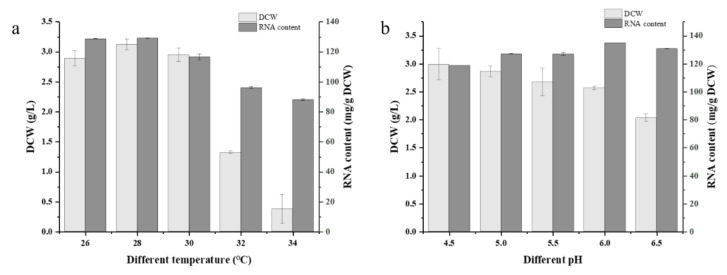
RNA content and growth in different culture temperature (**a**) and pH (**b**) before sterilization of G03H8. Data are the average of three independent experiments. Error bars represent ± SD.

**Figure 5 foods-11-02742-f005:**
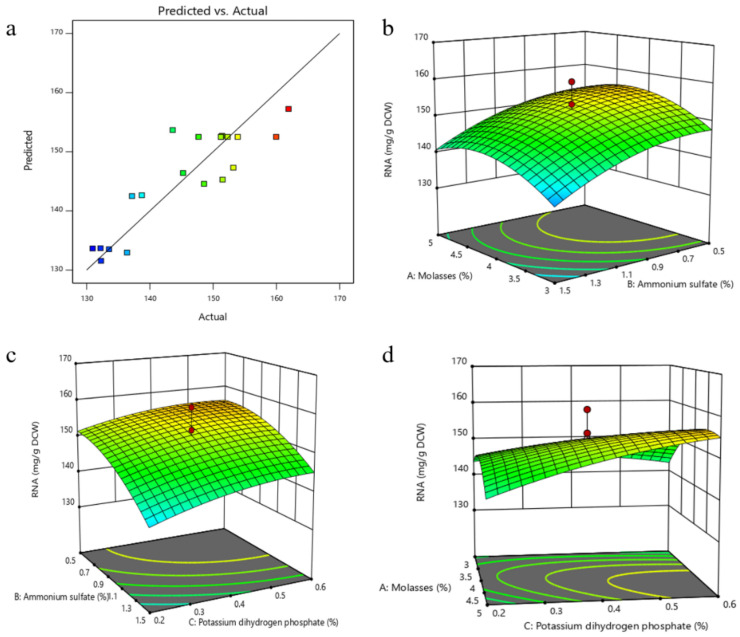
The result of response surface design (**a**), The relationship between the predicted value and the experimental value (Different colors reflected different ranges of RNA content, with blue to red getting higher); (**b**), Response surface curve as a function of molasses and (NH_4_)_2_SO_4_ concentrations when the KH_2_PO_4_ concentration was kept at 0.4%; (**c**), Response surface curve as a function of KH_2_PO_4_ and (NH_4_)_2_SO_4_ concentrations when the concentration of molasses was kept at 4%; (**d**), Response surface curve as a function of KH_2_PO_4_ and molasses concentrations when the (NH_4_)_2_SO_4_ concentration was kept at 1%).

**Figure 6 foods-11-02742-f006:**
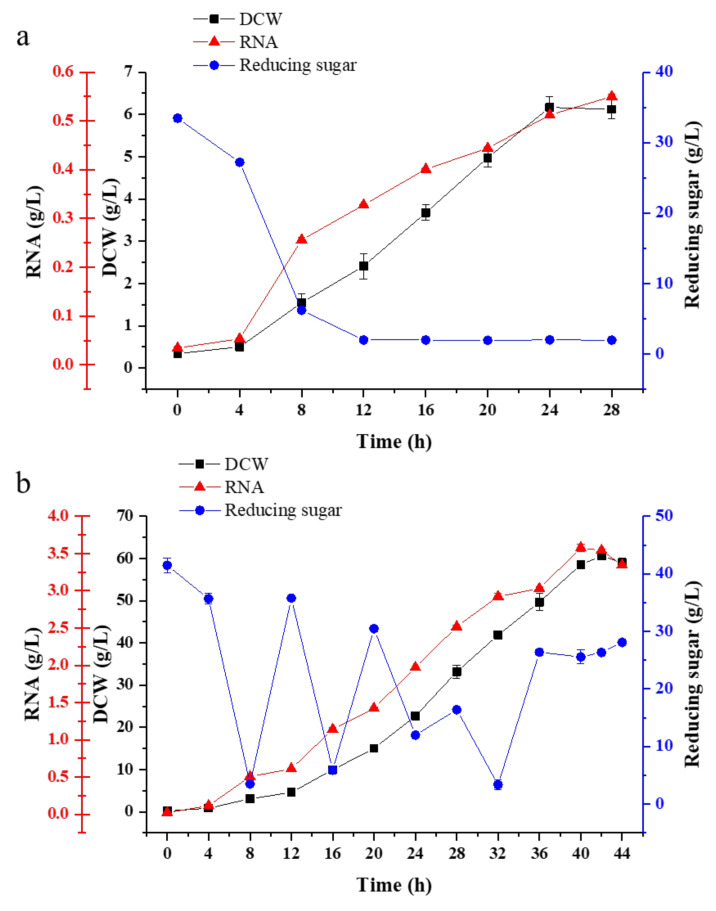
Production synthesis and changes in reducing sugar consumption of G03H8 in batch (**a**) and fed-bath (**b**) fermentation. Data are the average of three independent experiments. Error bars represent ± SD.

**Table 1 foods-11-02742-t001:** Levels of the factors tested in the PB design.

Factor	Level	
	−1	1
Molasses/%	2.00	4.00
(NH_4_)_2_SO_4_/%	1.00	3.00
MgSO_4_·7H_2_O/%	0.10	0.30
KH_2_PO_4_/%	0.10	0.30
FeSO_4_·7H_2_O/(g/L)	0.02	0.10
ZnSO_4_·7H_2_O/(g/L)	0.02	0.10
Yeast extract/%	0.50	1.50

**Table 2 foods-11-02742-t002:** Levels of the factors tested in the PB design.

Factor	Level				
	−1.68	−1	0	1	1.68
Molasses/%	2.32	3.0	4.0	5.0	5.68
(NH_4_)_2_SO_4_/%	0.16	0.5	1.0	1.5	1.84
KH_2_PO_4_/%	0.06	0.2	0.4	0.6	0.74

**Table 3 foods-11-02742-t003:** Results of flavored nucleotide content about G03 and G03H8.

	Regression Equation	R^2^	Retention Time/Min	Peak Area/mAU		DCW/mg		GMP + IMP/µg	
				G03	G03H8	G03	G03H8	G03	G03H8
GMP	y = 121,634x − 22,559	0.9997	6.16	2,001,359.50	2,752,181.00	2.51	3.14	88.53	178.46
IMP	y = 129,215x + 56,214	0.9992	7.26	3,626,890.00	8639122.50				

**Table 4 foods-11-02742-t004:** Regression results of the PB design.

Item	Coefficient	T-Value	*p*-Value
Constant	14.7740	357.41	0.000 *
Molasses/%	0.2191	5.30	0.006 *
(NH_4_)_2_SO_4_/%	−0.1424	−3.44	0.026 *
MgSO_4_·7H_2_O/%	−0.1015	−2.46	0.070
KH_2_PO_4_/%	0.1835	4.44	0.011 *
FeSO_4_·7H_2_O/(g/L)	−0.0757	−1.83	0.141
ZnSO_4_·7H_2_O/(g/L)	−0.0598	−1.45	0.222
Yeast extract/%	−0.0770	−1.86	0.136

* Statistically significant at 95% of confidence level.

**Table 5 foods-11-02742-t005:** Regression results of the central composite design.

Source	Sum of Squares	df	Mean Square	F-Value	*p*-Value
Model	13.4900	9	1.5000	4.1100	0.0190
A-Molasses	0.9769	1	0.9769	2.6800	0.1328
B-(NH_4_)_2_SO_4_	5.1700	1	5.1700	14.1800	0.0037
C-KH_2_PO_3_	1.2500	1	1.2500	3.4400	0.0934
AB	0.0000	1	0.0000	0.0001	0.9942
AC	1.0800	1	1.0800	2.9600	0.1163
BC	0.0425	1	0.0425	0.1164	0.7400
A^2^	3.7200	1	3.7200	10.1900	0.0096
B^2^	1.5300	1	1.5300	4.1800	0.0680
C^2^	0.4349	1	0.4349	1.1900	0.3005

**Table 6 foods-11-02742-t006:** Growth kinetic parameters of G03H8 cultivations in batch and fed-batch fermentation.

Fermentation Method	x (g/L)	µ (h^−1^)	Y_x/s_ (g/g)	r_x_ (g/L/h)	r_s_ (g/L/h)	Ef (%)
Batch	6.17	0.13	0.19	0.26	1.12	33.33
Fed-batch	60.58	0.12	0.43	1.58	3.20	75.44

x (g/L), the max biomass production, µ (h^−1^), the specific growth rate, r_x_ (g/L/h), the productivity or production rate of cell biomass, r_s_ (g/L/h), the utilization rate of substrate, Y_x/s_(g/g), the cell yield coefficient, Ef (%), the production efficiency.

## Data Availability

Data is contained within the article or Appendix A.

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
