# Peer review of "Fed-Batch Fermentation of Saccharomyces pastorianus with High Ribonucleic Acid Yield"

_foods, 2022, doi:10.3390/foods11182742_

Round 1
Reviewer 1 Report
The current work was motivated by the fact that Yeast is the major source for commercial Ribonucleic acid synthesis, and effective strain is the key to minimizing the production costs. A mutant S. pastorianus G03H8 with high RNA output was produced using ARTP mutagenesis. After response surface model improvement, the greatest RNA content of mutant strain G03H8 was found compared with the parental strain G03. These findings demonstrated that G03H8 has the potential to drastically lower the cost of RNA synthesis and shorten the fermentation cycle and gave useful knowledge on how to boost the RNA content in S. pastorianus strains for industrial production.
The research is excellent and well-written. However, small issues have been identified and require correction.
The quality of Figure 3, 4 and 5 should be improved
The quality of Figures s1, s4 and s5 should be improved
The following lines must be corrected and spell-checked. Line: 71, 109, 254, 282, 313
Author Response
Point 1:The quality of Figure 3, 4 and 5 should be improved
Response 1:The quality of all Figures had been improved and pixels met the requirements.
Point 2:The quality of Figures s1, s4 and s5 should be improved
Response 2: The quality of all Figures had been improved and pixels met the requirements.
Point 3:The following lines must be corrected and spell-checked. Line: 71, 109, 254, 282, 313
Response 3: The problems in these lines had been corrected.
Reviewer 2 Report
In the present study, the authors have developed a high RNA-yielding mutant of S. pastorianus named strain G03H8 using ARTP mutagenesis. Based on the response surface model optimization, the RNA content of the mutant increased by 40% from parental strain. The study seems to be well designed and conducted, the submitted manuscript almost satisfies the criteria. But I have only a minor comment about the manuscript as follows:
As the authors described in line 37, one of the brewer’s yeast S. pastorianus rather than Candida species is suitable to another source for RNA production on the viewpoint of its safety application. The authors gave an example a one of the RAN-producing strains C. tropicalis, which has been reportedly isolated from some neutropenic hosts. Allow me to ask the authors a primitive question, why C. tropicalis has been used in nowadays. Should the authors add information about some reasons with its priority on yields or handling?
Author Response
Point 1: Why C. tropicalis has been used in nowadays. Should the authors add information about some reasons with its priority on yields or handling?
Response 1: C. tropicalis has been used in nowadays because of its higher RNA content, but it has lower safety for food industry. I had added a reference for this in line 37.
Reviewer 3 Report
The manuscript entitled: Fed-batch fermentation of Saccharomyces pastorianus with high ribonucleic acid yield is very interesting. However in order to enhance the clarity and readability of the manuscript I suggest some changes/additions.
Ln40-41. Here some past work on genetic alterations are described. It would be very beneficial if the names of the genes and/or the metabolic pathway(s) are mentioned.
Ln42. The end of this sentence is incomplete. Fix the sentence and explain why GMO is a problem.
Ln45. Please provide an explanation of what ARTP is. There is more than enough "space" in the introduction to do so.
Ln47. That sentence makes little sense. How does that work?
Ln49. RSM does not improve productivity. It is an experimental/analytical approach. Please do not make it sound like any more than what it is.
Ln67. Where did your strain "G03" originally come from? Others must have the ability to reproduce your work, meaning that they should be able to get the same strain.
Ln 70. Please cite a reference that states that the addition of KCl helps in selecting mutants with high RNA yield.
Ln82. Define "normal saline"
Ln93. write vs not V.S.
Ln96. Since RNA and its measurement is key to this paper, I propose that a brief description of the analytical method is included.
Ln105. Remove the word "The" at the start of the sentence.
Never start a sentence with a number. Write the word in full. For instance One ml of ...
Ln113. provide a description on how you extracted RNA.
Table 1. please standardise the significance of the figures.
Lns 183-184. Remove hyphens from words.
Figure 2. Why don't you include the parent strain in the graph? Provide more information in the caption. For instance explain what the error bars mean.
Ln197. Write "twice as high" instead of "two times higher". It means the same, but it is better English
Figure 3 is nearly impossible to read. Please reproduce at higher resolution
Figure 4 is nearly impossible to read. Please reproduce at higher resolution
Figure 5. Please make the caption clearer to read.
Figure 6. Please reproduce at higher resolution
Author Response
Point 1: Ln40-41. Here some past work on genetic alterations are described. It would be very beneficial if the names of the genes and/or the metabolic pathway(s) are mentioned.
Response 1: I had mentioned the metabolic pathway or function of these genes.
Point 2: Ln42. The end of this sentence is incomplete. Fix the sentence and explain why GMO is a problem.
Response 2: The sentence had been fixed and the problem of GMO had been explained.
Point 3: Ln45. Please provide an explanation of what ARTP is. There is more than enough "space" in the introduction to do so.
Response 3: The ARTP had been explained in line 50.
Point 4: Ln47. That sentence makes little sense. How does that work?
Response 4: The sentence had been deleted and the screening method was explained in ‘2.2’.
Point 5: Ln49. RSM does not improve productivity. It is an experimental/analytical approach. Please do not make it sound like any more than what it is.
Response 5: The sentence had been corrected.
Point 6: Ln67. Where did your strain "G03" originally come from? Others must have the ability to reproduce your work, meaning that they should be able to get the same strain.
Response 6: The origin of G03 strain had been explained in ‘2.1’. It is presented by a brewery (Jiangsu, China) and a Frohberg type lager yeast strain used for industrial beer brewing.
Point 7: Ln 70. Please cite a reference that states that the addition of KCl helps in selecting mutants with high RNA yield.
Response 7: The reference had been cited.
Point 8: Ln82. Define "normal saline", Ln93. write vs not V.S.
Response 8: The "normal saline" had been defined and 'V.S' had been corrected.
Point 9: Ln96. Since RNA and its measurement is key to this paper, I propose that a brief description of the analytical method is included.
Response 9: A brief description of the RNA content measurement method had been added in ‘2.2’.
Point 10: Ln105. Remove the word "The" at the start of the sentence. Never start a sentence with a number. Write the word in full. For instance One ml of ...
Ln113. provide a description on how you extracted RNA.
Response 10: The problem had been corrected and RNA was extracted by method of RNA content measurement
Point 11: Table 1. please standardise the significance of the figures.
Response 11: The problem had been corrected.
Point 12: Lns 183-184. Remove hyphens from words. Ln197. Write "twice as high" instead of "two times higher". It means the same, but it is better English
Response 12: The problem had been corrected.
Point 13: Figure 2. Why don't you include the parent strain in the graph? Provide more information in the caption. For instance explain what the error bars mean.
Response 13: The parent strain had been added in Figure 2 and the error bars had been explained.
Point 14: Figure 3 is nearly impossible to read. Please reproduce at higher resolution. Figure 4 is nearly impossible to read. Please reproduce at higher resolution. Figure 5. Please make the caption clearer to read. Figure 6. Please reproduce at higher resolution
Response 14: The quality of all Figures had been improved and pixels met the requirements.
Reviewer 4 Report
This is a review of the article titled 'Fed-batch fermentation of Saccharomyces pastorianus with high ribonucleic acid yield'. The paper is well written. The issues pointed out below can improve the paper.
Line 29: Remove the comma after '[1]' and replace it with a full stop.
Line 58: In previous work. Proved the reference for previous work.
Line 60: Add 'In this study' before ARTP
Line 67: Why was S. pastorianus G03 used as the parent strain? What were the considerations?
Line 80: The specific modification carried out should be described to enable others to reproduce the work.
Line 83: How were the cells adjusted to 107 CFU/mL? OD or MacFarland standards?
Line 109: Remove the full stop after 'afterwards' and change 'And' to 'and'
Lines 153-154: Is there any reference to support that pH 6 is optimal?
Figure 3 and other figures: Can the axis labels be made darker?
References: This will need to be adjusted as per the guide.
The conclusion suggests that RNA generated from one strain can be used in other industrial production processes. Is there any limitation to this study considering that flavour or other desirable compounds may be strain-specific or differ among strains?
Author Response
Point 1: Line 29: Remove the comma after '[1]' and replace it with a full stop.
Response 1: The problem had been corrected.
Point 2: Line 58: In previous work. Proved the reference for previous work.
Response 2: The ‘previous work’ was an experiment that several yeast strains commonly used in food industry were evaluated for RNA production. The result had been added in supplementary materials.
Point 3: Line 60: Add 'In this study' before ARTP
Response 3: The problem had been corrected.
Point 4: Line 67: Why was S. pastorianus G03 used as the parent strain? What were the considerations?
Response 4: The reason had been explained in ‘2.1’.
Point 5: Line 80: The specific modification carried out should be described to enable others to reproduce the work.
Response 5: The specific modification had been carried out.
Point 6: Line 83: How were the cells adjusted to 107 CFU/mL? OD or MacFarland standards?
Response 6: The cells was adjusted to 107 CFU/ mL with normal saline by Cell Counter.
Point 7: Line 109: Remove the full stop after 'afterwards' and change 'And' to 'and'
Response 7: The problem had been corrected.
Point 8: Lines 153-154: Is there any reference to support that pH 6 is optimal?
Response 8: The pH 6 is optimal according the results of ‘Optimization of culture conditions’.
Point 9: Figure 3 and other figures: Can the axis labels be made darker?
Response 9: The quality of all Figures had been improved and the axis labels was darker.
Point 10: References: This will need to be adjusted as per the guide.
Response 10: The problem of some references had been corrected.
Point 11: The conclusion suggests that RNA generated from one strain can be used in other industrial production processes. Is there any limitation to this study considering that flavour or other desirable compounds may be strain-specific or differ among strains?
Response 11: RNA had been widely used in food, medicine, animal feeding and other industries. S. pastorianus is a generally recognized as safe (GRAS) microorganism. In China, the mutagenized safe strains are permitted to use in the food industry. Direct application of the strain may have some limitations, but RNA produced by the mutant G03H8 have no limitation for all these industries.
Round 2
Reviewer 3 Report
Most changes have been appropriately addressed. Some of the new text could do with some English language proof reading, some new language errors have been introduced.